# Tadalafil for Treatment of Fetal Growth Restriction: A Review of Experimental and Clinical Studies

**DOI:** 10.3390/biomedicines12040804

**Published:** 2024-04-04

**Authors:** Shintaro Maki, Sho Takakura, Makoto Tsuji, Shoichi Magawa, Yuya Tamaishi, Masafumi Nii, Michiko Kaneda, Kenta Yoshida, Kuniaki Toriyabe, Eiji Kondo, Tomoaki Ikeda

**Affiliations:** Department of Obstetrics and Gynecology, Mie University Graduate School of Medicine, Tsu 514-8507, Mie, Japan; s-takakura@med.mie-u.ac.jp (S.T.); tototo.9696tj@gmail.com (M.T.); shoichimagawa@med.mie-u.ac.jp (S.M.); tama8923@med.mie-u.ac.jp (Y.T.); m-nii1984@med.mie-u.ac.jp (M.N.); mkubo@med.mie-u.ac.jp (M.K.); yoshiken-obgy@med.mie-u.ac.jp (K.Y.); t-kuniaki@med.mie-u.ac.jp (K.T.); eijikon@med.mie-u.ac.jp (E.K.); t-ikeda@med.mie-u.ac.jp (T.I.)

**Keywords:** phosphodiesterase 5 inhibitor, tadalafil, fetal growth restriction

## Abstract

Fetal growth restriction (FGR) is a major concern in perinatal care. Various medications have been proposed as potential treatments for this serious condition. Nonetheless, there is still no definitive treatment. We studied tadalafil, a phosphodiesterase-5 inhibitor, as a therapeutic agent for FGR in clinical studies and animal experiments. In this review, we summarize our preclinical and clinical data on the use of tadalafil for FGR. Our studies in mouse models indicated that tadalafil improved FGR and hypertensive disorders of pregnancy. A phase II trial we conducted provided evidence supporting the efficacy of tadalafil in prolonging pregnancy (52.4 vs. 36.8 days; *p* = 0.03) and indicated a good safety profile for fetuses and neonates. Fetal, neonatal, and infant mortality was significantly lower in mothers receiving tadalafil treatment than that in controls (total number: 1 vs. 7, respectively; *p* = 0.03), and no severe adverse maternal events associated with tadalafil were observed. Although further studies are needed to establish the usefulness of tadalafil in FGR treatment, our research indicates that the use of tadalafil in FGR treatment may be a paradigm shift in perinatal care.

## 1. Introduction

Fetal growth restriction (FGR) is associated with perinatal mortality and morbidity [1,2,3,4] and can cause neurological sequelae in infants, such as cerebral palsy and pervasive developmental disorders [5,6]. Intrauterine hypoxia and malnutrition during fetal development may lead to cardiovascular disease and diabetes mellitus [7]. Furthermore, survival rates for babies born with severe FGR are low depending on the gestational age and severity [8,9], and the only way to manage this condition successfully is to deliver the fetus at the appropriate time.

Placental insufficiency is one of the causes of FGR. The placenta requires increased access to the maternal blood supply for fetal growth, which is achieved through extensive remodeling of the maternal spiral arteries, delivering blood directly into the placental intervillous space. Remodeling depends on the extravillous cytotrophoblasts, which invade the lining of the pregnant uterus at 18 weeks of gestation. In conditions of FGR, trophoblast invasion is inhibited, arteries are poorly remodeled, and the uteroplacental circulation capacity is extremely low [10]. As a result, fetal growth is suppressed. In cases of FGR associated with placental insufficiency, perinatologists have no definitive treatment options. The only strategy is to determine the optimal delivery time for the fetus.

Recently, phosphodiesterase 5 (PDE5) inhibitors have been investigated as promising medications for the treatment of FGR associated with placental insufficiency [11]. We studied tadalafil, a PDE5 inhibitor with a long half-life and high selectivity for PDE5, as a treatment option for FGR.

In this review, we summarize the results of our clinical and basic research on the use of tadalafil for the treatment of FGR.

## 2. Why Did We Select to Study Tadalafil as an Agent for the Treatment of FGR?

### 2.1. PDE5 Inhibitors

Inhibition of the enzymatic activity of PDE5 prevents inactivation of the intracellular second messenger cyclic guanosine monophosphate (cGMP) within vascular smooth muscle cells, which potentiates the activity of nitric oxide (NO), leading to vasodilatation and increased blood flow. Since PDE5 inhibitors also exert anti-inflammatory effects [12,13,14], and inflammation in the placenta is known to be a factor in the occurrence of placental insufficiency, treatment with a PDE5 inhibitor could be effective for placental insufficiency, including FGR [15].

Relatively common side effects include mild headache, flushing, dyspepsia, altered color vision, back pain and myalgias, and hypotension and dizziness, but these rarely lead to discontinuation of PDE5 inhibitors [16].

### 2.2. Sildenafil

Sildenafil is a PDE5 inhibitor that was first used in 1989 for erectile dysfunction [17]. Research on sildenafil citrate for the treatment of FGR has been ongoing, and the vasodilatory and vascular-endothelial-improving effects of sildenafil have been established in several studies [18,19,20,21,22,23]. Zoma et al. reported improved uterine blood flow via the cGMP-mediated endothelial relaxation of ovine uterine vessels in 2004, providing evidence suggesting its efficacy in improving fetoplacental perfusion in FGR [24]. Evidence from ex vivo and animal models of FGR indicate that sildenafil citrate increases birth weight and improves the uteroplacental blood flow [25,26,27].

In an ex vivo study using human specimens, Maharaj et al. demonstrated that sildenafil citrate acts as a vasodilator in human chorionic plate arteries [28]. In addition, it reportedly improves uterine blood flow via the cGMP-mediated endothelial relaxation of uterine vessels and causes relaxation of the human myometrium [24,29,30].

Several clinical studies have been conducted to treat FGR and pre-eclampsia (PE). Lin et al. reported that administering sildenafil citrate decreased the uterine pulsatility index and notching in FGR [31]. A randomized controlled trial reported the safe use of sildenafil during pregnancy and increased birth weights of neonates [32]. Additionally, it was reported that sildenafil was associated with increased fetal abdominal circumference growth velocity in severe early-onset FGR [33]. Based on the findings from these small clinical studies, the Sildenafil TheRapy in Dismal prognosis Early-onset intrauterine growth Restriction (STRIDER) consortium was launched. The study group evaluated sildenafil for the treatment of FGR with a prospective Individual Participant Data (IPD) and aggregate data systematic review with a meta-analysis and trial sequential analysis [34,35,36,37,38,39,40]. The STRIDER consortium, including the United Kingdom (UK), the Netherlands, New Zealand/Australia, and Canada, conducted randomized controlled trials examining the effects of sildenafil for the treatment of FGR [34]. Although these results were first reported in the UK, the primary outcome, the time from randomization to delivery, was not prolonged by sildenafil treatment for FGR [35]. Furthermore, in a randomized controlled trial (RCT) performed in New Zealand and Australia, the fetal growth velocity as a primary outcome was not increased by sildenafil [36]. The Dutch STRIDER group reported an increased risk of neonatal persistent pulmonary hypertension (PPHN) due to maternal sildenafil administration, which generated global media interest [37]. Based on the above, the benefits of sildenafil for the treatment of FGR are considered negative, and even harmful [11,41]. Subsequent data from neonates born to the Dutch STRIDER participants have been evaluated for causes of PPHN [40]. In that study, the authors speculated that sildenafil may have had a “rebound effect” on neonates due to sudden withdrawal of the drug at birth. Generally, sildenafil lowers blood pressure without directly affecting heart rate, but the analysis showed increased blood pressure in neonates. A similar phenomenon has been reported in human adults diagnosed with PPHN [42], classified as a “Type E” adverse drug reaction [43]. 

### 2.3. The Characteristics of Tadalafil

Tadalafil is a PDE5 inhibitor that acts as a vasodilator through the NO-cyclic guanosine monophosphate (cGMP) pathway and is prescribed for erectile dysfunction and PPHN [44,45]. Tadalafil reportedly improves vascular endothelial dysfunction and suppresses inflammation [46]. It is widely known that placental inflammation and endothelial dysfunction trigger PE; therefore, we can hypothesize that tadalafil would be effective against placental insufficiency [47,48]. Figure 1 summarizes the expected hypothetical properties of tadalafil. 

Tadalafil is a PDE5 inhibitor with several promising features: (1) it has a longer half-life and shows a more rapid onset of action than does sildenafil [49]; (2) food intake has negligible effects on its bioavailability [50]; and (3) it is particularly selective for the PDE5 enzyme and shows fewer side effects than sildenafil [51,52]. Tadalafil has been administered to vasodilate the pulmonary artery and improve pulmonary arterial hypertension (PAH). Eleven PDE isoforms (1–11) are found throughout the body. PDE5 is mainly expressed in the lungs and reproductive organs, including the placenta. A study on sheep reported that PDE5 is particularly highly expressed in the uterine artery during pregnancy and in the placenta and myometrium [53]. Therefore, it seems logical to attempt to improve uteroplacental circulation in cases of FGR using tadalafil, which is highly selective for PDE5. In our recent research, we demonstrated enhanced uterine blood flow with the use of tadalafil [54].

Our research on the use of tadalafil for the treatment of FGR was based on clinical experience. Women with severe PAH show high rates of small-for-gestational-age (SGA) infants [55]. However, it has been reported that when these women are treated with tadalafil, their infants were no longer SGA and were appropriate for gestational age [56].

The placental transport of tadalafil differs from that of sildenafil. Walton et al. [57] reported that sildenafil dilated the fetoplacental arteries and reduced fetal arterial pressure, whereas tadalafil had no effect on the fetus. One possible reason for this is that tadalafil does not cross the human placental barrier, nor does it have active metabolites. In our study, using an L-NG-nitroarginine methyl ester (L-NAME)-induced mouse model, tadalafil dilated the maternal placental blood sinuses, while no fetal capillary dilation was observed [58], which is believed to be due to the low placental transportability of tadalafil. From a pharmacological perspective, Nishimura et al. reported that in pregnant mice, the fetal transfer of tadalafil is limited due in part to a difference in the contribution of breast cancer resistance protein (BCRP) to fetus-to-mother transport across the placenta [59]. These results also indicate that tadalafil might act less on the fetal side.

Our previous report suggested that tadalafil restored the levels of hypoxia-inducible factor-2α (HIF-2α), phospho-rpS6, and eukaryotic translation initiation factor-4E (eIF-4E) in the placenta of women with FGR, which act downstream of the mammalian target of rapamycin (mTOR) signaling that regulates oxygen and nutrition [60]. mTOR signaling is important for the growth of fetal organs. Therefore, tadalafil may treat FGR by improving oxygenation and nutrition rather than by directly increasing blood flow of the fetus. Tadalafil has the potential to improve FGR not through direct effects on the fetus, but through improvements in uteroplacental perfusion and nutritional exchange at the placental barrier.

### 2.4. Case Report

This case study was initiated in 2014 after obtaining permission from the Institutional Review Board [61]. Our first experience treating FGR with tadalafil was that of a 41-year-old primigravida Japanese woman who became pregnant after in vitro fertilization pre-embryo transfer. At 22 weeks and 4 days of gestation, the estimated fetal weight (EFW) was 309 g (–2.6 standard deviation [SD]), and severe oligohydramnios was observed. After informed consent for this treatment was obtained from the patient and her husband, she was administered a 20 mg/day tablet of tadalafil. The fetal bladder started to dilate 4 days after starting tadalafil, and the volume of amniotic fluid increased at 10 days. The weight of the fetus increased by approximately 50–100 g per week. The patient did not experience adverse events. Because the amount of amniotic fluid had decreased and the fetal heart rate monitoring showed many variable decelerations at 32 weeks of gestation, we performed a cesarean section. A 1024 g baby was delivered with Apgar scores of 5 and 7 at 1 and 5 min after birth, respectively. At 5 years of age, the baby boy had no neurodevelopmental abnormalities, and his height and weight reached the reference range. 

### 2.5. Case–Control Study

We also conducted a case–control study to assess maternal and perinatal outcomes of tadalafil treatment in pregnant women with FGR [62]. Eleven pregnant women treated with tadalafil with FGR were compared with fourteen pregnant women matched for maternal age, parity, gestational age, and estimated fetal weight at enrollment and treated with conventional treatment according to the existing guidelines for obstetric practice in Japan [63]. The fetal growth velocity (FGV) from enrollment to delivery was significantly higher in the tadalafil group (median: 17.7 g/day; interquartile range (IQR): 10.6–23.0 g/day) than in the conventional management group (median: 12.8 g/day; IQR: 0–17.2 g/day). Based on these results, we considered FGV to be the most important outcome and decided to use it as the primary outcome in subsequent clinical trials.

### 2.6. Phase I Trial

A phase I trial was conducted to investigate the safety of tadalafil administration to the mother for FGR [64]. According to the case–cohort method, the dose was started at 10 mg/day and sequentially increased to 20 and 40 mg/day while evaluating the onset of adverse events. Adverse events were evaluated according to the National Cancer Institute Common Terminology Criteria for Adverse Events [65]. No serious adverse events were causally related to the maternal administration of tadalafil. The only fatal adverse event was intrauterine death, which was believed to be caused by velamentous insertion of the umbilical cord and was determined by the Safety Evaluation Committee to have no causal relationship with tadalafil. No adverse neonatal events related to tadalafil administration were observed. 

### 2.7. Phase II Trial

Next, a phase II multicenter, randomized, controlled trial (TADAlafil treatment for fetuses with early-onset growth restriction: TADAFER II) was conducted at a major medical center in Japan in September 2016 [66]. This study investigated the efficacy and safety of tadalafil in the treatment of FGR using an open-label design. Singleton pregnant women with FGR, with a gestation period ranging from 22 weeks and 0 days to 33 weeks and 6 days, were randomized into two groups, and the primary outcome was FGV. One group received only conventional treatment for FGR according to the guidelines for obstetric practice in Japan [63] (conventional treatment group), and the other group received 20 mg/day of tadalafil in conjunction with conventional treatment (tadalafil treatment group). 

Following the results of STRIDER UK, which reported no benefits of sildenafil in pregnancy prolongation [35], the Japan Agency for Medical Research and Development concluded that all PDE5 inhibitors were ineffective against FGR due to the results and recommended that we stop recruiting new candidates for the TADAFER II trial at the end of March 2018. A total of 89 patients (tadalafil treatment group, 45 patients; conventional treatment group, 44 patients) who participated at that time were analyzed in this study. For the safety assessment, fetal, neonatal, and infant deaths were compared between the tadalafil and conventional treatment groups. There were seven cases (four fetal deaths, one neonatal death, and two infant deaths) in the conventional treatment group and only one case (neonatal death) in the tadalafil treatment group (Figure 2). 

Two cases of REDV and AEDV improved. This finding is reasonable because tadalafil ameliorates placental dysfunction (Table 1).

The rate of neonatal adverse events did not differ between the two groups, with two cases of PPHN occurring in each group [66]. This is an important finding in the aspect of fetal safety because PPHN has been reported worldwide in the STRIDER trial [41]. Among the adverse maternal events, headache and facial flushing were more common in the tadalafil group; however, no severe adverse events associated with tadalafil were observed. In particular, while no significant differences were found in FGV, which is the primary outcome, prolongation of pregnancy was observed in the tadalafil treatment group in early-onset FGR (Figure 3). 

Grade 1 headache, facial flushing, and nasal hemorrhage, which had no influence on maternal quality of life, were frequently observed in the tadalafil treatment group.

Additional analysis of the TADAFER II data showed that tadalafil decreased the umbilical artery pulsatility index when the EFW was less than −2.0 SD (unpublished data). This indicates that vascular resistance to blood flow in the uteroplacental and fetoplacental circulations is reduced, which was also reported on the study of sildenafil [67,68].

TADAFER II was an incomplete study; therefore, we were unable to prove the efficacy of tadalafil, and further studies are required.

### 2.8. Basic Research

We investigated the efficacy of tadalafil in an L-NAME-induced mouse model of PE with FGR [58]. Pregnant C57BL/6 mice were divided into two groups: a control group of dams received 0.5% carboxymethylcellulose (CMC) dissolved in drinking water (C-dams), and the L-NAME group received 1 mg/mL L-NAME dissolved in 0.5% CMC, which was further divided into the tadalafil-treated group (TL-dam) and L-NAME-only group (L-dam). Blood pressure, proteinuria, and fetal weight of PE and FGR mice were improved in TL-dams compared to L-dams. In TL-dams, placental maternal blood sinuses were significantly narrower than those in C-dams, and the placental growth factor (PLGF) concentration was increased. Interestingly, fetal capillaries did not show a change in TL-dams in contrast to dilated maternal blood sinuses, which means that tadalafil has less transportability in the placenta.

Two studies have verified the efficacy of tadalafil [69,70]. Li et al. reported that the prophylactic administration of tadalafil reduced hypertension, proteinuria, FGR, and flow-mediated dilatation; balanced endothelial-relative factors; and prevented the activation of inflammation in the placenta and kidney tissues in rat models with induced PE [69]. Lambert et at. reported that administering tadalafil lowered tumor necrosis factor (TNF)-α, interleukin (IL)-6, and keratinocyte-derived chemokine/growth-related oncogene (KC/GRO) levels in plasma; additionally, it reduced TNF-α expression in the placenta of halogen-inhalation-induced pulmonary and systemic injuries in pregnant mice [70]. A report suggests that tadalafil suppresses inflammatory cytokines [71], which highlights its potential efficacy in treating placental insufficiency.

Subsequently, we studied the neuroprotective action of tadalafil in an L-NAME-induced mouse model [72]. The number of HIF-2α-positive cells in the white matter and dentate gyrus regions of the fetal hippocampus was higher for the L-dams than for the C-dams but was significantly lower for the TL-dams than for the L-dams. In the labyrinth zone of the placenta, the number of HIF-2α+ cells was also significantly lower in the TL-dams than in the L-dams. The number of cells positive for glial fibrillary acid protein (GFAP) in the corpus callosum was significantly lower in the TL offspring than in the L offspring. The area positive for myelin basic protein (MBP) in the cingulum was significantly improved in the TL offspring compared to that in the L offspring. Since poor neurodevelopment is a serious problem in FGR [73], this result may shed light on improving neurologic prognoses in humans.

We also investigated the efficacy of tadalafil in another mouse model of FGR: the reduced uterine perfusion pressure (RUPP) model. To establish the RUPP model, bilateral uterine and ovarian vessels were ligated with a nylon thread (0.75 mm in diameter), and the thread was immediately removed, leaving a small space [74]. In the RUPP model, elevated maternal sBP and FGR levels also improved. The sFlt-1 concentration was significantly decreased in the placenta of the RUPP group treated with tadalafil. Furthermore, tadalafil treatment improves renal pathologies such as endotheliosis and increases the mesangial area [75].

### 2.9. Developmental Prognosis of Children Treated with Tadalafil

We also evaluated the development prognosis of children in the long term [76]. The developmental quotient (DQ) was evaluated using the Kyoto Scale of Psychological Development (KSPD) test at 1.5 years of corrected age, 3 years old. The median score of each area (postural–motor functions, P–M; cognitive–adaptive functions, C–A; and language–social functions, L–S) in the tadalafil treatment group exceeded 85 (normal range), and the proportion of cases that had a DQ ≤ 70 was less than 20%, which was considered favorable. Although this study was conducted in a single institution and evaluated a single group of children treated with tadalafil, it provides valuable data for further research.

### 2.10. TADAFER IIb

A multicenter, placebo-controlled, randomized controlled trial (TADAFER IIb), which started in 2018, is currently ongoing [77]. Since TADAFER II was an incomplete study and had an open-label design that did not use a placebo, further trials are necessary to establish a higher level of evidence.

The study population is as follows: (1) pregnant women aged ≥20 and <45 years; (2) EFW of −1.5 SD or less of the mean EFW for gestational age (GA) according to the Japanese standard curve [78]; (3) GA between 20 + 0 and 31 + 6 weeks; (4) expected date of confinement determined using the criteria of the guidelines for obstetrical practice in Japan (2017) [79]; (5) singleton pregnant women; and (6) provision of signed written informed consent from the pregnant women.

The patient will be randomly assigned to the following three arms.

Arm A: Patients will receive a placebo twice daily along with conventional treatment until delivery.

Arm B: Patients will receive 20 mg of tadalafil once per day and placebo once per day (morning: tadalafil, evening: placebo) in conjunction with conventional treatment until delivery.

Arm C: Patients will receive 20 mg of tadalafil twice daily (total 40 mg) along with conventional management until delivery.

Since the previous study (TADAFER II) was terminated midway, and the TADAFER IIb protocol includes the verification of the effects of a dose of 40 mg/day (20 mg × 2) in addition to the dose of 20 mg/day of tadalafil, this study was designated as an “exploratory study” phase II trial. The blood concentration of tadalafil decreases over time, as the time to maximum concentration is 4 h [80] (Figure 4). Considering that maintaining blood concentration throughout the day is necessary to maintain blood flow, we designed an arm for oral administration twice daily.

The primary endpoint is the prolongation of GA, defined as the days from the first day of protocol-defined treatment to birth, which was significantly prolonged in the tadalafil treatment group in the TADAFER II study [66]. The case registration period was originally set to end in May 2022. However, as the planned number of cases was not achieved, it has been extended to May 2024.

## 3. Systematic Reviews

A systematic review of the maternal and perinatal safety and clinical outcomes of PDE5 inhibitors was published in 2022 [81]. The review determined that tadalafil was safe for pregnant women, fetuses, and neonates. Lim et al. described an analysis of new candidates for the prevention and treatment of FGR [11]. Tadalafil was ranked as a “medium-potential” drug for the treatment of FGR. Given that sildenafil has been associated with increased neonatal PPHN, caution regarding the safe use of tadalafil was included in the article. The safety profile and distinctions from sildenafil have been outlined above.

In 2023, Pels et al. published a systematic review of FGR treatments active via the NO pathway [82]. The review reported that interventions affecting the NO pathway did not influence fetal and neonatal mortality in pregnant women with FGR. However, there are limited data available on tadalafil, and its supporting evidence is classified as “low”. As mentioned earlier, tadalafil and sildenafil exhibit distinct effects, suggesting potential variations in the effectiveness of tadalafil. Further data are required to provide a more comprehensive understanding of the role of tadalafil for the treatment of FGR.

## 4. Conclusions and Future Aspects

Tadalafil is a promising therapeutic agent for FGR that can improve placental function in terms of uteroplacental perfusion and nutritional exchange. Research on tadalafil has gradually advanced, and to date no other therapeutic argent has received greater research interest for the treatment of FGR, which makes it a practical option. Although the present study focused on treatment for FGR, in the future we will also investigate tadalafil from the perspective of a preventive drug for placental insufficiency. A placebo-controlled, randomized controlled trial, TADAFER IIb, is being conducted; case registration is ongoing, and we await the results.

## Figures and Tables

**Figure 1 biomedicines-12-00804-f001:**
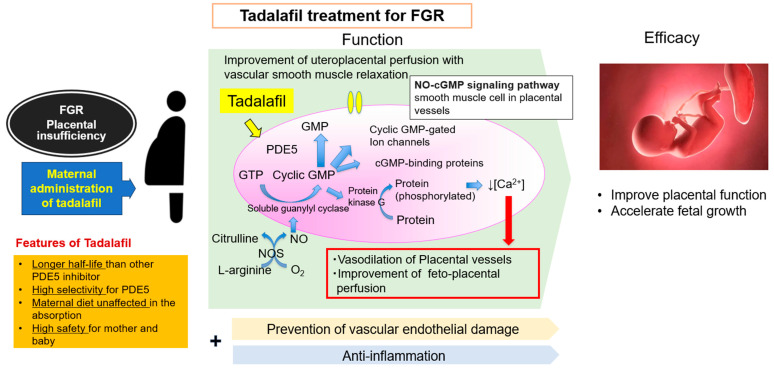
Summary of hypotheses and characteristics regarding the effects of tadalafil. FGR: fetal growth restriction, PDE5: phosphodiesterase 5, GMP: guanosine monophosphate, NO: nitric oxide, NOS: nitric oxide synthase.

**Figure 2 biomedicines-12-00804-f002:**
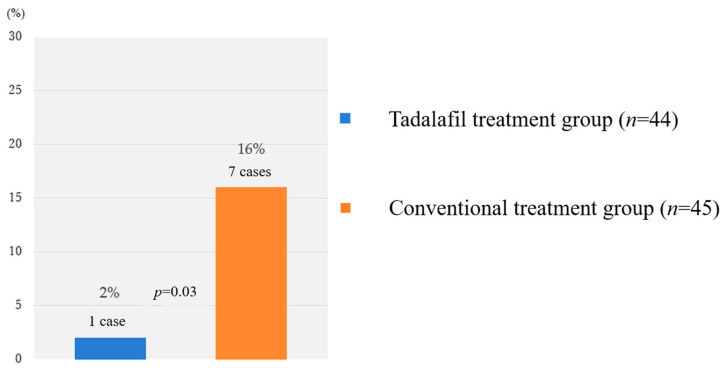
Total number of fetal, neonatal, and infant death cases in TADAFER II study.

**Figure 3 biomedicines-12-00804-f003:**
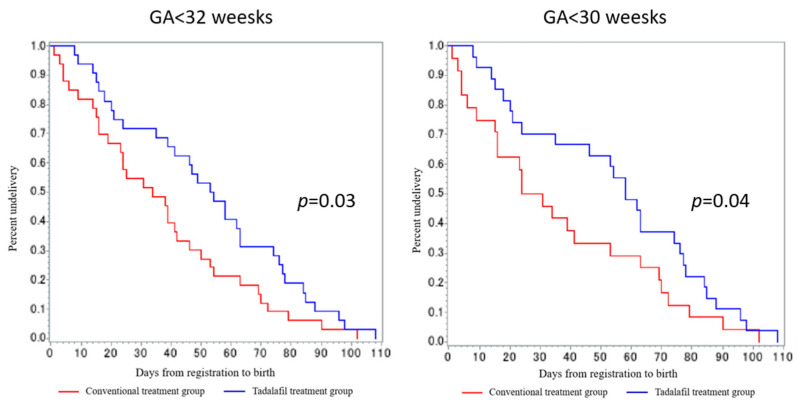
Prolongation of gestational weeks. Prolongation of GA was defined as days from the first day of protocol-defined treatment to birth. The figure above shows the prolongation of GA in Kaplan–Meier curves for each GA at treatment initiation (<32 weeks, <30 weeks), comparing binary data across two groups using the generalized Wilcoxon test.

**Figure 4 biomedicines-12-00804-f004:**
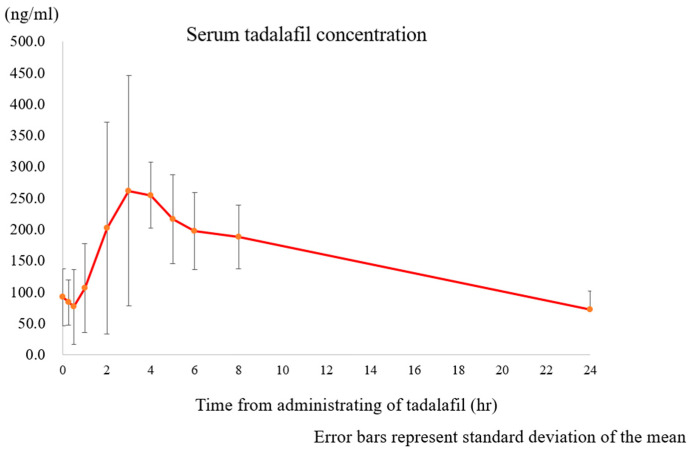
Serum concentration of tadalafil in pregnancy.

**Table 1 biomedicines-12-00804-t001:** Details of death cases (TADAFER II).

Case	Allocation	GW at Registration (Week)	BW (g)	UA REDV or AEDV
Fetal death	Conventional treatment group	23	328	+
Fetal death	Conventional treatment group	20	<300	+
Fetal death	Conventional treatment group	25	440	+
Fetal death	Conventional treatment group	21	484	−
Neonatal death	Tadalafil treatment group	21	317	−
Neonatal death	Conventional treatment group	24	440	−
Infant death	Conventional treatment group	27	704	−
Infant death	Conventional treatment group	28	730	−

GW: gestational weeks. BW: birth weight. UA: umbilical artery. REDV: reversed end-diastolic velocity. AEDV: absent end-diastolic velocity.

## Data Availability

Not applicable.

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
