# Peer review of "Tadalafil for Treatment of Fetal Growth Restriction: A Review of Experimental and Clinical Studies"

_biomedicines, 2024, doi:10.3390/biomedicines12040804_

Round 1

Reviewer 1 Report

Comments and Suggestions for Authors

In this review, ,, Tadalafil for Treatment of Fetal Growth Restriction: A Review 2 of Experimental and Clinical Studies,, by Shintaro Maki et al., the authors summarized the results of clinical and basic research on the use of tadalafil for the treatment of fetal growth restriction (FGR).

Comments and Suggestions for Authors

The article has some shortcomings:

Authors should double-check abbreviations and make the necessary corrections so that abbreviations are explained when they first appear, both in the abstract and in the manuscript text and figure legends.

Line 81 – delete ,, . ”

Line  100, 129, 136,  137, – explaine abbreviations

Line 214 - after explaining the abbreviation in line 89, use the abbreviation

Line 234-235 - after explaining the abbreviation in line 129, use the abbreviation

Section Conclusions – The Conclusions section needs to be revised, it is too long; Lines 321-334 should be moved from this section

Reviewer 2 Report

Comments and Suggestions for Authors

The manuscript “`Tadalafil for treatment of fetal growth restriction: A review of experimental and clinical studies” gives new information about the important function of Tadalafil as a therapeutic reagent for Fetal Growth Restriction (FGR).

I think the manuscript is well-documented and also believe that the accumulation of results from research like this manuscript provides a great contribution to clinical treatment in the future. It needs an additional description and scheme of the author’s results to make it easier for the reader to read.

Major points

1. Additional information is needed about the side effects of therapeutic agents, including Tadalafil.

2. Does Tadalafil only work on inhibiting PDE5? Is any possible that other effects of Tadalafil are improving FGR?

3. Please describe any improvements needed to increase the number of cases using Tadalafil treatment in the future.

3. In this study, the authors focused on the effective functions of Tadalafil for FGR. It is better to add a schematic model of the relationship among potential factors (NO, blood flow, PDE, and so on) and which agents including Tafafil are more effective for FGR (because Tadalafil is medium potential).

Reviewer 3 Report

Comments and Suggestions for Authors

The review “Tadalafil for Treatment of Fetal Growth Restriction: A Review of Experimental and Clinical Studies” addresses an interesting topic. I have following comments/ Suggestions,

1.      The abstract does not give any idea about the  numbers ( deaths etc.) and it will be interesting for the readers if some aggregated data is presented here.

2.      The results are well described.

3.      Figure 4, it is not clear the error bars are around mean or median?

4.      Conclusion is clear and also provides insight into the previously published reviews on similar topic. 

Reviewer 4 Report

Comments and Suggestions for Authors

Thank you for the opportunity to review the manuscript entitled 'Tadalafil for Treatment of Fetal Growth Restriction: A Review of Experimental and Clinical Studies'. I would recommend this manuscript for publication. The study is a synthesis of all studies on the effect of tadalafin on fetal growth and it is based predominantly on the work of the present research group. Interestingly, the European studies provide different results than those in Japan, but it seems that Tadalafil instead of sildenafil will be the key substance in the FGR treatment. I have just two minor remarks:

side 7: 'Several reports have verified' --> Two studies ... (there was cited only 2 studies)

In the next paragraph:

typographic mistake:  'fetal hippocampal hippocampus was higher' --hippocampal is unnecessary.  

Round 2

Reviewer 2 Report

Comments and Suggestions for Authors

I think that the revised manuscript has been improved.